# Study protocol for COVID-RV: a multicentre prospective observational cohort study of right ventricular dysfunction in ventilated patients with COVID-19

Jennifer Mary Willder [1] Philip McCall,[2,3] Claudia-Martina Messow,[4] Mike Gillies,[5,6] Colin Berry,[7,8] Benjamin Shelley[2,3]

For numbered affiliations see end of article.

**Correspondence to**
Dr Benjamin Shelley;
Benjamin.Shelley@glasgow.ac.uk

## ABSTRACT

**Introduction** COVID-19 can cause severe acute respiratory failure requiring management in intensive care unit with invasive ventilation and a 40% mortality rate. Cardiovascular manifestations are common and studies have shown an increase in right ventricular (RV) dysfunction associated with mortality. These studies, however, comprise heterogeneous patient groups with few requiring invasive ventilation. This study will investigate the prevalence and prognostic significance of RV dysfunction in ventilated patients with COVID-19 which may lead to targeted interventions to improve patient outcomes.

**Methods and analysis** This prospective multicentre observational cohort study will perform transthoracic echocardiography (TTE) in 150 patients with COVID-19 requiring invasive ventilation for more than 48 hours. RV dysfunction will be defined as TTE evidence of RV dilatation along with the presence of septal flattening. Baseline demographics, disease severity data and clinical information relating to proposed aetiological mechanisms of RV dysfunction (acute respiratory distress syndrome (ARDS), disordered coagulation, direct myocardial injury and ventilation) will be collected and analysed. Primary outcome measures include the prevalence of RV dysfunction and its association with 30-day mortality. Exploratory outcome measures will investigate the association of the proposed aetiological mechanisms of RV dysfunction to the primary outcomes. Prevalence of RV dysfunction will be determined along with 95% Clopper-Pearson CIs and 30-day survival will be analysed using logistic regression adjusting for patient demographics, phase of disease and baseline severity of illness. The role of potential aetiological factors (ARDS, disordered coagulation, direct myocardial injury and ventilation) in relation to the primary outcomes will be analysed using logistic regression.

**Ethics and dissemination** Approval was gained from Scotland A Research Ethics Committee (REC reference 20/SS/0059). Findings will be disseminated by various methods including webinars, international presentations and publication in peer-reviewed journals.

### Strengths and limitations of this study

► Includes the most severely ill patients with COVID-19 disease; a patient group which are under-represented in other studies of right ventricular (RV) function in patients with COVID-19.

► Pragmatic study and primary outcome, reflecting clinical practice of intensive care unit echocardiography.

► Appropriately statistically powered.

► RV function is dynamic.

► As with all COVID-19 research, recruitment of adequate participant numbers may be hampered due to the 'passing of the peak' of the COVID-19 pandemic in the UK but new cases continue to emerge.

## INTRODUCTION

In December 2019, the first case of a novel COVID-19 caused by the SARS-CoV-2 virus was identified in Wuhan, China.[1] Although the majority of cases are mild, approximately 5% of confirmed cases will require intensive care unit (ICU) admission, with severe acute respiratory failure (SARF) being the prominent feature.[2] As of 27 September 2020 there were over 32.7 million cases worldwide with over 990 000 confirmed deaths.[3] On 19 June 2020, over 11 500 patients had been treated in UK critical care units.[4 5] Unfortunately, in line with other countries, UK figures have demonstrated extremely poor ICU outcomes with mortality rates of around 40%.[4]

Although not always classical in presentation, the acute respiratory distress syndrome (ARDS) is widely accepted to be the usual cause of SARF in patients with COVID-19.[6] Along with respiratory symptoms, cardiovascular manifestations are also common and may adversely impact prognosis.[7–9] In ARDS, the more susceptible right ventricle (RV)

**BMJ**

is subject to impaired function due to increased after-load.[10 11] RV dysfunction is known to be an independent predictor of mortality in the non-COVID-19 setting and increasing ARDS severity is associated with increased frequency of RV dysfunction.[12–15]

Recent studies have shown that RV dysfunction is a common finding in patients with COVID-19 and it increases mortality.[9 16 17] While no single causative factor has been elucidated, there are several potential aetiological factors which have been shown to increase RV afterload and impair RV contractility, with both factors leading to RV dysfunction. ARDS,[10 18 19] pulmonary microthrombosis and macrothrombosis,[20–24] and positive pressure ventilation with positive end-expiratory pressure[18 25] have all been shown to increase RV afterload with RV contractility being depressed by direct myocardial injury.[7 26 27] Previous studies addressing RV dysfunction in patients with COVID-19 included heterogeneous groups of patients of varying disease severity, of whom only a small proportion required mechanical ventilation. No adequately powered study has prospectively explored the incidence of RV dysfunction and its association with mortality in ventilated patients with COVID-19. Further investigation and exploration of causative mechanisms may allow targeted interventions to improve outcomes for this patient group.[12 19 28 29]

### Hypothesis

Right ventricular failure is common in patients ventilated with COVID-19 and is associated with increased mortality. In examining this hypothesis, we will explore potential mechanisms for RV dysfunction in this patient group.

## METHODS AND ANALYSIS

Summary: A multicentre prospective observational study in ventilated patients with COVID-19 in participating Scottish ICUs.

Centre(s): 15 participating ICUs in Scotland.

Study status: This is a planned study that was granted ethical approval on 5 June 2020 and grant funding on 27 May 2020. Recruitment commenced in September 2020.

### Selection of study subjects

#### Inclusion criteria

Patients >16 years with confirmed SARS-CoV-2 infection with SARF requiring tracheal intubation and positive pressure ventilation in intensive care for more than 48 hours, but not more than 14 days will be eligible for inclusion in the study.

#### Exclusion criteria

Patients for whom informed consent is unable to be provided either directly or on their behalf by a relative/welfare guardian will be excluded. Other exclusion criteria include:
► Pregnancy.

► Ongoing participation in any investigational research that may undermine the scientific basis of the study.
► Prior participation in the COVID-RV study.
► Requirement for extracorporeal membrane oxygenation support for respiratory or cardiovascular failure.
► Patient at end of life and not expected to survive longer than 24 hours.

### Study conduct

#### Recruitment

Potential participants will be identified and recruited by the local treating clinical team during their admission to intensive care.

#### Consent

Prospective informed consent will be obtained. As target patients will be critically ill in the ICU, it is unlikely that they themselves will be able to provide informed consent. Consent will be sought from the patient's legally designated representative, most likely a relative, either in person or via telephone. Further consent will then be sought from the study participants if they recover sufficiently and regain capacity to provide consent.

#### Medical management

Medical management will be according to the standard of care at each treating site and is not influenced by this study protocol.

#### Study interventions

*Echocardiography conduct*

Participants recruited to take part in the study will undergo a single transthoracic echocardiogram (TTE) to determine the presence or absence of RV dysfunction. This will be undertaken after 48 hours but before 14 days of intubation and invasive ventilation. TTE will be performed by a range of appropriately competent practitioners including (but not exclusively) intensive care clinicians, cardiologists and specialist echocardiographers reflecting the clinical practice of bedside echocardiography in intensive care. For the purposes of the primary outcome, imaging required will be in keeping with the protocol required for a focused intensive care echo scan.[30] A secondary quantitative data set will be collected where echocardiography skills allow.

*Laboratory sampling*

High sensitivity troponin (I or T) and, where available, natriuretic peptides (brain natriuretic peptide (BNP) or N-terminal pro b-type natriuretic peptide (NT-proBNP)) will be measured as part of routine blood sampling on the same day as TTE.

#### Data collection and management

Data collection will be performed by the local study team on case report forms (CRFs) which will be filed and securely stored at participating sites. The data will be anonymised at site and a unique alphanumeric study number allocated. Completed CRFs will then be entered

onto a secure online database in a linked anonymised form. Electronic data will be stored in an encrypted and anonymised format for 10 years following the completion of the trial. At the end of this period, the data set will be destroyed and this will be performed to DoD 5220.22-M standards. The study will adhere to all data protection regulations and all data will be held in accordance with the General Data Protection Regulation (2018).

### Clinical data

Clinical data involve baseline demographic data and chronic comorbidities, ICU admission information, severity of illness and acute comorbidities and follow-up data. Clinical data relating to the proposed mechanisms of RV dysfunction will also be collected, specifically regarding ARDS, disordered coagulation, myocardial injury and ventilation (table 1).

### Laboratory data

Laboratory data will be obtained from the local biochemistry and haematology laboratory reporting systems on the day of echocardiography and at follow-up.

### ECG data

A 12-lead ECG will be anonymised and transferred electronically to the research team for further analysis. Abnormalities in rate, rhythm, conduction and repolarisation (ST-segment deviation, T-wave inversion) will be classified, blind to other clinical data.

### Echocardiography data

A focused echocardiography data set will be used to answer the primary outcome (table 2). If available, and the echocardiographer's competency and experience permit, further quantitative measures of RV function will be obtained at this time (table 3).

As part of the data set, a parasternal short axis and apical four-chamber (A4C) 'loop' will be recorded and saved. For the A4C loop this will take the form of an RV-focused view. For both, the images will be optimised, have ECG monitoring present throughout and the frame rate should be maximised; both will be recorded for four beats.[31] These data will be anonymised and transferred electronically to the research team for further analysis.

## Study outcomes

### Primary outcome

The primary outcome of the study is the prevalence of RV dysfunction in ventilated patients with COVID-19 and its association with 30-day mortality. RV dysfunction will be defined as TTE evidence of RV dilatation along with the presence of septal flattening (in systole, diastole or both). RV dilatation will be determined from the A4C view at end diastole and will be defined as when the RV:LV ratio is >1:1.

### Justification for primary outcome

TTE is the mainstay of RV imaging in ICU; it is non-invasive, free from ionising radiation, widely available and low cost. The ability to bring the echo machine to the patient bedside

**Table 1** Clinical and laboratory data to explore mechanisms of RV dysfunction

| Mechanistic category | Clinical and laboratory data |
|---|---|
| ARDS | Requirement for prone ventilation |
| | Murray lung injury score and components |
| | Requirement for paralysis |
| | Compliance |
| | Arterial blood gas analysis |
| | Referral for extracorporeal membrane oxygenation (ECMO) |
| Disordered coagulation | D-dimer |
| | Systemic anticoagulation (prophylactic/therapeutic) |
| | Confirmed or suspected pulmonary thromboembolism |
| | Prothrombin time (PT)/activated partial thromboplastin time (aPTT) |
| | Platelet count |
| Myocardial injury | LV function (from TTE) |
| | $ScvO_2$ |
| | Natriuretic peptides (BNP or NT-proBNP) |
| | Troponin |
| | Acute coronary syndrome treatment |
| | ECG changes |
| | Inotropic/vasopressor support |
| | History/treatment of arrhythmia |
| Ventilation | Fraction of inspired oxygen ($FiO_2$) |
| | Ventilatory mode |
| | Positive end-expiratory pressure |
| | Peak airway pressure |
| | Driving pressure |
| | Respiratory rate |

Clinical and laboratory data to explore mechanisms of RV dysfunction in each mechanistic category: ARDS, disordered coagulation, myocardial injury and ventilation. ARDS, acute respiratory distress syndrome; BNP, brain natriuretic peptide; LV, left ventricle; NT-proBNP, N-terminal pro b-type natriuretic peptide; RV, right ventricle; TTE, transthoracic echocardiogram.

means it is the preferred method for diagnosis of RV dysfunction in ICU patients.[32] TTE can determine RV dilatation and can detect eccentric movement of the interventricular septum. The combination of RV dilatation and eccentric septal motion is often referred to as acute cor pulmonale and is a widely reported measure of RV dysfunction in patients with respiratory failure in intensive care.[13 14 33–37]

### Exploratory outcomes

Exploratory outcome measures largely relate to the association of the proposed aetiological factors, namely

**Table 2** Focused echocardiography data set relating to primary outcome

| Echocardiography parameter | Result | |
|---|---|---|
| RV dilatation | Yes | |
| | No | |
| | Unable to assess | |
| Septal flattening | Yes | Systole |
| | | Diastole |
| | | Both |
| | No | |
| | Unable to assess | |
| RV function | Normal | |
| | RV dysfunction | |
| | Unable to assess | |
| Free text: *Comments* | | |
| LV function | Normal | |
| | LV dysfunction | |
| | Unable to assess | |
| LV dilatation | Normal | |
| | LV dilatation | |
| | Unable to assess | |
| Free text: *Comments* | | |

Basic echocardiography data set to allow determination of the primary outcome measures.
LV, left ventricle; RV, right ventricle.

**Table 3** Detailed echocardiography measures of RV function

| Echocardiography parameter | Result |
|---|---|
| RV area | |
| RV end diastolic area | $cm^2$ |
| RV end systolic area | $cm^2$ |
| Fractional area change (FAC) | % |
| | Unable to assess |
| Free text: *Image quality, RWMA, McConnell's sign, presence of clot, and so on* | |
| Tricuspid annular plane systolic excursion (TAPSE) | mm |
| | Unable to assess |
| S' wave velocity at tricuspid annulus (S' Wave) | cm/s |
| | Unable to assess |
| RV dimensions | |
| RVD1 | mm |
| RVD2 | mm |
| RVD3 | mm |
| | Unable to assess |
| TR $V_{max}$ (to allow calculation of PA systolic pressure) | m/s |
| | Unable to assess |
| Tricuspid regurgitation | None |
| | Mild |
| | Moderate |
| | Severe |
| | Unable to assess |
| Free text: *Method of assessing TR (vena contracta, quantitative (EROA, ERV), hepatic vein flow)* | |
| Right ventricular index of myocardial performance (RIMP)/Tei index | |
| | PWD |
| | Tissue Doppler |
| | Unable to assess |
| IVC diameter | |
| IVC diameter | cm |
| Respiratory variation/collapse >50% | Yes |
| | No |
| | Unable to assess |
| RVOT acceleration time | ms |
| | Unable to assess |
| LV function and size | |
| 2D | |
| LVIDd | cm |
| LVIDs | cm |
| LVFS | % |
| | Unable to assess |
| Biplane | |
| LVEDV | mL |

Continued

ARDS, disordered coagulation, direct myocardial injury and ventilation, to RV dysfunction and 30-day mortality. The association of cardiac biomarkers (troponin T or I and natriuretic peptides (BNP or NT-proBNP)) to the primary outcome measures will also be evaluated.

Saved echo loops will be analysed centrally at the host institution, blinded and anonymised under the supervision of British Society of Echocardiography (BSE)-accredited echocardiographers. In addition to exploring the utility of other echocardiographic parameters such as eccentricity index and the tricuspid annular plane systolic excursion/pulmonary artery systolic pressure ratio, 2D speckle tracking will be performed for analysis for RV free wall and global longitudinal strain. Image analysis will be performed on the TomTec 2D Cardiac Performance Analysis platform. Agreement on classification of echocardiographic parameters between critical care echocardiographers at each study site and echocardiographers at the host centre will be explored.

## Statistical considerations

All statistical analyses will be performed in conjunction with the Robertson Centre for Biostatistics at the University of Glasgow.

**Table 3** Continued

| Echocardiography parameter | Result |
|---|---|
| LVESV | mL |
| LVEF | % |
| | Unable to assess |

*Free text: Comments such as regional wall motion abnormality, valve pathology, and so on*

Detailed echocardiography measures of RV function to be undertaken dependent on the echocardiographer's competency and experience and if image quality allows. EROA, estimated regurgitant orifice area; ERV, effective regurgitant volume; IVC, inferior vena cava; LV, left ventricle; LVEDV, left ventricular end-diastolic volume; LVEF, left ventricular ejection fraction; LVESV, left ventricular end-systolic volume; LVFS, left ventricular fractional shortening; LVIDd, left ventricular internal diameter diastolic; LVIDs, left ventricular internal diameter systolic; PA, pulmonary artery; PWD, pulse wave doppler; RV, right ventricle; RVOT, right ventricular outflow tract; RWMA, regional wall motion abnormality; TR, tricuspid regurgitation.

## Sample size calculation

Sample size selection is required to be a pragmatic balance of maximising available information versus the prompt delivery of the study and reporting of results to the clinical workforce. Given the number of patients admitted to ICU in Scotland during the early stages of the pandemic, it is realistic to recruit 120–150 patients across participating sites.[5]

Prevalence of RV dysfunction: With a sample size of 120, if the proportion of patients with RV dysfunction was 50%, the 95% CI around the estimated rate would be 40.7% to 59.3%, if the proportion was 25% the 95% CI would be 17.5% to 33.7%. With a sample size of 150 patients, the 95% CIs would be (41.7 to 58.3) and (18.6 to 33.1), respectively.

Association with mortality: Table 4 demonstrates the differences in mortality rates that could be detected in a

sample size of 120–150 patients at a significance level of 5% and proportions of patients with RV dysfunction of 25% and 50%, assuming an overall mortality rate of 50% (as presented in the recent UK Intensive Care National Audit & Research Centre report on COVID-19 in critical care).[38]

## Data analysis
### *Primary outcome*
The proportion of ventilated patients with COVID-19 who have RV dysfunction will be determined, with a 95% CI (Clopper-Pearson method). The association of RV dysfunction with 30-day mortality will then be analysed using logistic regression analysis predicting 30-day ICU mortality from presence or absence of RV dysfunction, adjusting for patient demographics, phase of disease and baseline severity of illness.

### *Exploratory outcomes*
To assess the role of potential aetiological factors (ARDS, disordered coagulation, direct myocardial injury and ventilation) for RV dysfunction, the relation of each measure of the aetiological factors to the outcomes will be analysed in the same way as for the primary outcome. A more in-depth analysis will include carrying out a principal component analysis for all measures within each factor, in order to create one single measure (the first principal component) representing each factor. The relation of these to the outcomes will be assessed in regression analyses analogously to the ones described above. This will be done for each factor separately and also in combination. Other approaches, for example, the least absolute shrinkage and selection operator, for determining the role of the individual measures of all factors may be explored.[39]

The association of cardiac biomarkers with RV dysfunction will be analysed using logistic regression analysis predicting presence or absence of RV dysfunction from the biomarker level, adjusting for patient demographics, phase of disease and baseline severity. Association of the

**Table 4** Indicative power

| Sample size | Power (%) | Number *without* RV dysfunction | Number *with* RV dysfunction | Mortality in patients *without* RV dysfunction | Mortality in patients *with* RV dysfunction | OR |
|---|---|---|---|---|---|---|
| 120 | 80 | 60 | 60 | 0.373 | 0.627 | 2.83 |
| 120 | 90 | 60 | 60 | 0.354 | 0.646 | 3.33 |
| 150 | 80 | 75 | 75 | 0.386 | 0.614 | 2.53 |
| 150 | 90 | 75 | 75 | 0.369 | 0.631 | 2.92 |
| 120 | 80 | 90 | 30 | 0.427 | 0.719 | 3.43 |
| 120 | 90 | 90 | 30 | 0.417 | 0.749 | 4.17 |
| 150 | 80 | 112 | 38 | 0.435 | 0.695 | 2.96 |
| 150 | 90 | 112 | 38 | 0.426 | 0.722 | 3.50 |

Differences in mortality rates that could be detected at a 5% significance level for sample sizes of 120 and 150, and proportions of patients with RV dysfunction of 25% and 50%, assuming an overall mortality rate of 50%.
RV, right ventricle.

biomarkers with 30-day mortality will be analysed in a similar manner. To assess whether any association differs in the presence or absence of RV dysfunction, RV dysfunction and its interaction with the biomarker will be added to the model.

## Patient and public involvement

Due to the nature of the current COVID-19 pandemic and the subsequent urgency to establish this study there was no patient or public involvement. Patients were not invited to comment on the study design and were not consulted to develop patient-relevant outcomes or interpret the results. Patients were not invited to contribute to the writing or editing of this document for readability or accuracy.

## ETHICS AND DISSEMINATION

Ethical approval was granted by the Scotland A Research Ethics Committee (REC reference 20/SS/0059) on 5 June 2020.

The safety profile of all research interventions (TTE, collection of clinical information and blood specimens) is well established and therefore adverse event reporting is not applicable.

Given the nature of the current global pandemic we will endeavour to publish the prevalence of RV dysfunction in COVID-19 as soon as possible on completion of the study. Once the 30-day follow-up period is complete, the primary outcome will be published along with all available exploratory analyses. We envisage sharing findings with the critical care community at an early stage through a series of webinars.

## Author affiliations
[1]West of Scotland School of Anaesthesia, NHS Education for Scotland West Region, Glasgow, UK
[2]Academic Unit of Anaesthesia, Pain and Critical Care Medicine, University of Glasgow, Glasgow, UK
[3]Department of Anaesthesia, Golden Jubilee Hospital, Clydebank, UK
[4]Robertson Centre for Biostatistics, University of Glasgow, Glasgow, UK
[5]Anaesthesia, Care and Pain Medicine, The University of Edinburgh, Edinburgh, UK
[6]Department of Anaesthesia, Edinburgh Royal Infirmary, Edinburgh, UK
[7]Department of Cardiology, Golden Jubilee Hospital, Clydebank, UK
[8]British Heart Foundation Glasgow Cardiovascular Research Centre, University of Glasgow, Glasgow, UK

**Contributors** BS and PM conceived the study and BS is the grant holder. MG and CB assisted in the initial study design. JMW wrote the patient documentation and developed the case report forms and online data collection database. CMM provided guidance on sample size requirements and statistical analysis and oversaw the statistical analyses. All authors contributed to refinement of the study protocol and approved the final manuscript.

**Funding** This work was supported by Medical Research Scotland (grant number CVG-1730-2020). CB is supported by British Heart Foundation Centre of Research Excellence grant (reference number RE/18/6/34217).

**Competing interests** BS reports personal fees from Janssen-Cilag, outside the submitted work.

**Patient and public involvement** Patients and/or the public were not involved in the design, or conduct, or reporting, or dissemination plans of this research.

**Patient consent for publication** Not required.

**Provenance and peer review** Not commissioned; externally peer reviewed.

**ORCID iD**
Jennifer Mary Willder http://orcid.org/0000-0003-4843-8717

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
