## [Reviewer comments · BMJ Open]

ARTICLE DETAILS

TITLE (PROVISIONAL)	Study protocol for COVID-RV: a multi-centre prospective observational cohort study of right ventricular dysfunction in ventilated patients with COVID-19.
AUTHORS	Willder, Jennifer; McCall, Philip; Messow, Martina; Gillies, Mike; Berry, Colin; Shelley, Benjamin

VERSION 1 – REVIEW

REVIEWER	Salvatore Grasso University of Bari "Aldo" Moro", Bari, ITALY
REVIEW RETURNED	26-Jul-2020

GENERAL COMMENTS	The study protocol by Willder and coworkers has the purpose of defining the prevalence of right ventricular (RV) dysfunction in mechanically ventilated patient with COVID-19 induced ARDS. Overall, this is a pragmatic study, reflecting the ICU practice of assessing the RV function by transthoracic echo at the bedside. As such, more complex and likely difficult to obtain, invasive assessments of RV function are not included. I appreciated this approach and the rigorous methodological and statistical approach, considering also the objective difficulties of studying patients with COVID 19 in the ICU context
--

REVIEWER	Juan C Grignola Department of Pathophysiology Facultad de Medicina Universidad de la República Montevideo-URUGUAY
REVIEW RETURNED	16-Aug-2020

GENERAL COMMENTS	I congratulate the authors for their exciting and well-proposed study protocol. As the authors claimed, the different studies on cardiovascular complications in COVID-19 patients are heterogeneous both in illness severity and type of CV manifestation studied. In ICU patients, the RV is susceptible to dysfunction and, in turn, RV dysfunction causes and aggravates common critical diseases. Acute RV afterload increase (ARDS, pulmonary microthrombosis, mechanical ventilation) and depress contractility (direct myocardial injury) present in severe infection with COVID-19 configures the perfect storm to produce RV dysfunction/failure leading to increase mortality. The present study protocol proposes to explore the incidence of RV dysfunction and its association with mortality in ventilated patients with COVID-19 prospectively. Additionally, the authors will
---

	explore the causal mechanisms that could be involved in RV dysfunction. I have some methodological points that should be clarified and some items to be added to improve the study: It would be advisable to add the definitions of severe acute respiratory failure (SARF), ARDS, and right ventricular dysfunction (RV dilatation: operating cut-off values of RV dimensions). Specify which clinical and laboratory data will be obtained, particularly relating to RV dysfunction proposed mechanisms. Regarding the echocardiographic data to obtain, it is advisable to include the linear and area measurements to assess RV dilation relative to the left ventricle (RV/LV end-diastolic diameter, RV/LV end-diastolic area), LV eccentricity index and RV wall thickness. The last one would allow differentiating acute versus acute-on-chronic RV dysfunction (Krishnan S & Schmidt GA. Chest 2015; 147:835-46. Huang SJ et al. Int Care Med 2018; 44:868-83). For example: acute cor pulmonale was defined by the association of RV dilatation in the long-axis view of the heart (RVEDA/LVEDA >0.6) and a visually identified systolic paradoxical ventricular septal motion in the short-axis view of the heart. Finally, the authors could add the TAPSE/SPAP ratio estimation as a RV-pulmonary coupling index. Considering that the prevalence of patent foramen ovale can reach 25%, do the authors plan to assess the presence of PFO? (Legras, A., Caille, A., Begot, E. et al. Acute respiratory distress syndrome (ARDS)-associated acute cor pulmonale and patent foramen ovale: a multicenter noninvasive hemodynamic study. Crit Care 2015; 19:174). Minor points: In page 7, third paragraph, line 3: replace by: "...which have been shown to increase RV afterload including ARDS, pulmonary microthrombosis and positive pressure ventilation and to depress RV contractility by direct myocardial injury, both factors leading to RV dysfunction. On page 14, last sentence: "...e.g., the lasso, for determining...." What is the meaning of the lasso? Add in bibliography:  • Krishnan S & Schmidt GA. Acute right ventricular dysfunction. Real-time management with echocardiography. Chest 2015; 147:835-46. (excellent review). • Huang SJ et al. The use of echocardiographic indices in defining and assessing right ventricular systolic function in critical care research. Int Care Med 2018; 44:868-83.
--	--

REVIEWER	Luu Maxime CHU Dijon Clinical investigation Center - clinical trial unit University Hospital of Dijon, Dijon France
REVIEW RETURNED	14-Sep-2020

GENERAL COMMENTS	This protocol is clearly presented and of importance regarding the current epidemiologic situation and the frequency of RV dysfunction in patients ventilated with COVID. One main limit in my opinion is the absence of insurance of TTE standardized evaluation by experimented investigators. although understandable in this situation, authors should try their best to have dedicated investigators for TTE evaluation. Other Comments:  - history of RV and recent or iterative ARDS (anteriority to define) should be taken in account in the analysis, if not in the exclusion criteria. this can definitely induce heterogeneity in included population and impact prognosis evaluation. Sample size calculation: Due to the unprecedented situation, even if not classic, i agree with the authors to choose a pragmatic presentation of their sample size calculation as a range estimation. Principal Component Analysis can be relevant here, but no method to avoid model overfitting is described, which seems in my opinion important here. Minor comments  - p.13 : "sample size selection...[...] to inform care" to be removed. this generic sentence do not add relevant information on the protocol - p.13: "we believe..." should be reserved for discussion, not in methods. Patient inclusion feasibility should be based on prior data when possible, even if coming from other areas.
---

VERSION 1 – AUTHOR RESPONSE

Reviewer(s)' Comments to Author:

Reviewer: 1

Reviewer Name: Salvatore Grasso

Institution and Country: University of Bari "Aldo" Moro", Bari, ITALY

Competing interests: None declared

Please leave your comments for the authors below

The study protocol by Willder and coworkers has the purpose of defining the prevalence of right ventricular (RV) dysfunction in mechanically ventilated patient with COVID-19 induced ARDS. Overall, this is a pragmatic study, reflecting the ICU practice of assessing the RV function by transthoracic echo at the bedside. As such, more complex and likely difficult to obtain, invasive assessments of RV function are not included. I appreciated this approach and the rigorous methodological and statistical approach, considering also the objective difficulties of studying patients with COVID 19 in the ICU context

Thank you. We hope a pragmatic, real world, approach allows us to provide an answer to this challenging question.

Reviewer: 2

BMJ Open uses compulsory open peer review. Your name and institution will be returned to the authors and will be published with this review if the article is accepted.

By submitting your review you agree to the peer review terms and conditions. If the article is published, your name and review will also be published as supplementary information to the article. This means the review will be made available under the same Creative Commons license granted to the article.

Reviewer Name: Juan C Grignola
Institution and Country:
Department of Pathophysiology
Facultad de Medicina
Universidad de la República
Montevideo-URUGUAY
Competing interests: None declared

Please leave your comments for the authors below

I congratulate the authors for their exciting and well-proposed study protocol.

As the authors claimed, the different studies on cardiovascular complications in COVID-19 patients are heterogeneous both in illness severity and type of CV manifestation studied.

In ICU patients, the RV is susceptible to dysfunction and, in turn, RV dysfunction causes and aggravates common critical diseases. Acute RV afterload increase (ARDS, pulmonary microthrombosis, mechanical ventilation) and depress contractility (direct myocardial injury) present in severe infection with COVID-19 configures the perfect storm to produce RV dysfunction/failure leading to increase mortality.

The present study protocol proposes to explore the incidence of RV dysfunction and its association with mortality in ventilated patients with COVID-19 prospectively. Additionally, the authors will explore the causal mechanisms that could be involved in RV dysfunction.

I have some methodological points that should be clarified and some items to be added to improve the study:

It would be advisable to add the definitions of severe acute respiratory failure (SARF), ARDS, and right ventricular dysfunction (RV dilatation: operating cut-off values of RV dimensions).

In keeping with the pragmatic nature of this study, we have chosen to define respiratory failure as the need tracheal intubation and intermittent positive pressure ventilation. This method also allows us to explore if the severity of respiratory failure (P:F ratio, requirement for prone ventilation etc) is important when exploring mechanisms of RV dysfunction.

RV dilatation will be defined in this study when the RV:LV ratio is more than 1:1 on the apical 4 chamber view. A comment to this effect has been added under the "echocardiography data" section of the manuscript.

Specify which clinical and laboratory data will be obtained, particularly relating to RV dysfunction proposed mechanisms.

This has been added as a new table 1.

Regarding the echocardiographic data to obtain, it is advisable to include the linear and area measurements to assess RV dilation relative to the left ventricle (RV/LV end-diastolic diameter, RV/LV end-diastolic area), LV eccentricity index and RV wall thickness.

As clarified above, the ratio of RV to LV size at end diastole will be used to determine RV dilatation. Where the echocardiographer's competency and experience permit additional, quantitative, parameters will be collected as part of the "detailed" echocardiography assessment (Table 3). Additionally, images will be transferred to research team at the host institution for further analysis on an exploratory basis. Although not explicitly stated in the manuscript (where we previously only mentioned speckle tracked strain), we would have measured eccentricity index and RV free wall thickness. The TAPSE/SPAP ratio mentioned later in this review also looks very interesting and is a parameter we will be exploring. The manuscript has been amended accordingly. In conceiving this study, we felt it was important that it could be delivered by the critical care clinicians treating these patients. Although 'ideal' from a research perspective, we didn't want to overburden them with an echocardiography protocol that they would be unable to obtain, particularly in a potentially overburdened healthcare system. This is why the echo protocol is pragmatic. We hope to tease out the utility of some of these more esoteric parameters as part of any exploratory analysis of echo images. The last one would allow differentiating acute versus acute-on-chronic RV dysfunction (Krishnan S & Schmidt GA. *Chest* 2015; 147:835-46. Huang SJ et al. *Int Care Med* 2018; 44:868-83).

For example: acute cor pulmonale was defined by the association of RV dilatation in the long-axis view of the heart ($RVEDA/LVEDA > 0.6$) and a visually identified systolic paradoxical ventricular septal motion in the short-axis view of the heart.

Finally, the authors could add the TAPSE/SPAP ratio estimation as a RV-pulmonary coupling index. Considering that the prevalence of patent foramen ovale can reach 25%, do the authors plan to assess the presence of PFO?

The TAPSE/SPAP ratio looks very interesting and something we will definitely look to explore as part of a secondary analysis of any echo results.

In keeping with the response above, we do not plan to prospectively assess for the presence of PFO. (Legras, A., Caille, A., Begot, E. et al. Acute respiratory distress syndrome (ARDS)-associated acute cor pulmonale and patent foramen ovale: a multicenter noninvasive hemodynamic study. *Crit Care* 2015; 19:174).

Minor points:

In page 7, third paragraph, line 3: replace by: "...which have been shown to increase RV afterload including ARDS, pulmonary microthrombosis and positive pressure ventilation and to depress RV contractility by direct myocardial injury, both factors leading to RV dysfunction.

Thank you, this has been changed to separate the role of RV afterload and RV contractility in RV dysfunction.

On page 14, last sentence: "...e.g., the lasso, for determining....." What is the meaning of the lasso? The lasso (least absolute shrinkage and selection operator) is a regression analysis method that minimises the residual sum of squares subject to the sum of the absolute coefficients being less than a constant. I have added a reference in the manuscript to this effect.

Add in bibliography:

- Krishnan S & Schmidt GA. Acute right ventricular dysfunction. Real-time management with echocardiography. *Chest* 2015; 147:835-46. (excellent review).

- Huang SJ et al. The use of echocardiographic indices in defining and assessing right ventricular systolic function in critical care research. *Int Care Med* 2018; 44:868-83.

Thank-you, both of these references have been added.

Reviewer: 3

Reviewer Name: Luu Maxime

Institution and Country:

CHU Dijon

Clinical investigation Center - clinical trial unit

University Hospital of Dijon, Dijon

France

Competing interests: None declared

Please leave your comments for the authors below

This protocol is clearly presented and of importance regarding the current epidemiologic situation and the frequency of RV dysfunction in patients ventilated with COVID. One main limit in my opinion is the absence of insurance of TTE standardized evaluation by experimented investigators. although understandable in this situation, authors should try their best to have dedicated investigators for TTE evaluation.

We agree wholeheartedly, that in the ideal research setting, standardised evaluation of TTE images by a core team would lead to robust results.

However, in designing this study we wanted to create a protocol that could be delivered by critical care clinicians at the bedside. These are the doctors that are seeing these patients and depending on the results of this study will be implementing interventions based on its findings. Also, in the current pandemic, we feel this design is the best way to obtain results that we can promptly return to clinicians.

The parameters used as part of the primary outcome (RV dilatation and septal dyskinesia) are all those that would be obtained as part of a basic critical care (Faculty of Intensive Care Echo [FICE]) echo study. The majority of doctors performing TTE in this study will have at least FICE accreditation and all will be supported by their department in performing echo as part of their normal clinical practice.

We are asking for images to be transferred to ourselves for secondary analysis. This is to explore other parameters such as speckle tracked strain. As part of this further analysis, we will be exploring agreement between critical care echocardiographers and the 'central' BSE accredited echocardiographers for the primary outcome. A comment to this effect has been added under the exploratory outcomes heading.

Other Comments:

- history of RV and recent or iterative ARDS (anteriority to define) should be taken in account in the analysis, if not in the exclusion criteria. this can definitely induce heterogeneity in included population and impact prognosis evaluation.

The severity of ARDS and the timing of echocardiography in relation to the onset of respiratory failure will be accounted for within the analysis.

Although not ideal, we did not put a history of RV dysfunction as an exclusion criterion as we felt this may be a barrier to participation. If clinicians felt that participation was dependent on knowing the status of RV function prior to hospital or ICU admission, they may not have been felt suitable for recruitment. This was a pragmatic approach in the current pandemic setting.

As highlighted by another reviewer, we will explore the role of RV free wall hypertrophy as a potential method to differentiate between acute and acute-on-chronic RV dysfunction.

Sample size calculation: Due to the unprecedented situation, even if not classic, i agree with the authors to choose a pragmatic presentation of their sample size calculation as a range estimation.

Principal Component Analysis can be relevant here, but no method to avoid model overfitting is described, which seems in my opinion important here.

Thank you, we feel the pragmatic approach to sample size calculation is a strength of this study. Particularly with the uncertainty of the ongoing pandemic.

Our statistician responds that principal component analysis does not necessarily lead to overfitting and can in fact also lead to underfitting, as each component is created so that it captures maximal variability, but not maximal relation to the outcome of interest. It is possible that the information relevant to the outcome is partly lost in this step. Since we're planning to use the first principal component only in the regression analysis, overfitting should not be an issue. We will of course also look at the variance components when interpreting the results. Since there is no optimal way of analysing this kind of data without more prior knowledge, we're planning to also do other exploratory analyses using different methods (e.g. the lasso, as mentioned in the text) and hope that, by looking at these results in conjunction, we can get a clearer picture of the role of the different aetiological factors.

We have not altered the manuscript to this effect.

Minor comments

- p.13 : "sample size selection...[...] to inform care" to be removed. this generic sentence do not add relevant information on the protocol

This has been removed.

- p.13: "we believe..." should be reserved for discussion, not in methods. Patient inclusion feasibility should be based on prior data when possible, even if coming from other areas.

This has been amended

VERSION 2 – REVIEW

REVIEWER	Juan C Grignola Department of Pathophysiology Facultad de Medicina. Hospital de Clínicas. Universidad de la República Montevideo-URUGUAY
REVIEW RETURNED	14-Oct-2020

GENERAL COMMENTS	The authors have responded to previously made observations.
---

REVIEWER	Maxime Luu Clinical Investigation Center- clinical trial unit University Hospital of Dijon, Dijon, FRANCE
REVIEW RETURNED	29-Oct-2020

GENERAL COMMENTS	I thank the authors for clearly addressing all my comments. I have no further questions regarding this protocol and wish them good luck for the investigation phase of the trial. - I still feel omission of RV dysfunction history in exclusion criteria may impair the strength of your results, but can be accepted in a pragmatic trial. I suggest authors to raise this point in their future publication of results. - Combination of variance checking + sensitivity analyses + lasso is a basic way to check this potential issue, but not optimal. it is still acceptable if conducted by an experienced statistician.
---